

# Comprehensive analysis and identification of key genes and signaling pathways in the occurrence and metastasis of cutaneous melanoma

Hanying Dai, Lihuang Guo, Mingyue Lin, Zhenbo Cheng, Jiancheng Li, Jinxia Tang, Xisha Huan, Yue Huang and Keqian Xu

Department of Laboratory Medicine, the Third Xiangya Hospital, Central South University, ChangSha, HuNan, People's Republic of China
Department of Laboratory Medicine, Xiangya School of Medicine, Central South University, ChangSha, HuNan, People's Republic of China

Corresponding author
Keqian Xu, xukeqian@csu.edu.cn

## ABSTRACT

**Background**. Melanoma is a malignant tumor of melanocytes, and the incidence has increased faster than any other cancer over the past half century. Most primary melanoma can be cured by local excision, but metastatic melanoma has a poor prognosis. Cutaneous melanoma (CM) is prone to metastasis, so the research on the mechanism of melanoma occurrence and metastasis will be beneficial to diagnose early, improve treatment, and prolong life survival. In this study, we compared the gene expression of normal skin (N), primary cutaneous melanoma (PM) and metastatic cutaneous melanoma (MM) in the Gene Expression Omnibus (GEO) database. Then we identified the key genes and molecular pathways that may be involved in the development and metastasis of cutaneous melanoma, thus to discover potential markers or therapeutic targets.

**Methods**. Three gene expression profiles (GSE7553, GSE15605 and GSE46517) were downloaded from the GEO database, which contained 225 tissue samples. R software identified the differentially expressed genes (DEGs) between pairs of N, PM and MM samples in the three sets of data. Subsequently, we analyzed the gene ontology (GO) function and Kyoto Encyclopedia of Genes and Genomes (KEGG) pathway of the DEGs, and constructed a protein-protein interaction (PPI) network. MCODE was used to seek the most important modules in PPI network, and then the GO function and KEGG pathway of them were analyzed. Finally, the hub genes were calculated by the cytoHubba in Cytoscape software. The Cancer Genome Atlas (TCGA) data were analyzed using UALCAN and GEPIA to validate the hub genes and analyze the prognosis of patients.

**Results**. A total of 134, 317 and 147 DEGs were identified between N, PM and MM in pair. GO functions and KEGG pathways analysis results showed that the upregulated DEGs mainly concentrated in cell division, spindle microtubule, protein kinase activity and the pathway of transcriptional misregulation in cancer. The downregulated DEGs occurred in epidermis development, extracellular exosome, structural molecule activity, metabolic pathways and p53 signaling pathway. The PPI network obtained the most important module, whose GO function and KEGG pathway were enriched in oxidoreductase activity, cell division, cell exosomes, protein binding, structural molecule activity, and metabolic pathways. 14, 18 and 18 DEGs were identified

respectively as the hub genes between N, PM and MM, and TCGA data confirmed the expression differences of hub genes. In addition, the overall survival curve of hub genes showed that the differences in these genes may lead to a significant decrease in overall survival of melanoma patients.

**Conclusions.** In this study, several hub genes were found from normal skin, primary melanoma and metastatic melanoma samples. These hub genes may play an important role in the production, invasion, recurrence or death of CM, and may provide new ideas and potential targets for its diagnosis or treatment.

## INTRODUCTION

Cutaneous melanoma (CM) is the most dangerous type of skin cancer. It accounts for approximately 232,100 new cases of CM around the world each year, including 55,500 deaths (*Schadendorf et al., 2018*), and it ranks 15th among the most common cancers in the world (*Leonardi et al., 2018*). CM is one of the most aggressive and metastatic human cancers, and compared with other cancer types, it can spread from a small primary tumor to multiple sites throughout the body (*Bostel et al., 2016*). Although primary cutaneous melanoma can be removed and cured through the operation, when a few millimeters thick skin lesion is found, it represents an advanced stage, and there is a high chance of distant visceral metastasis (*Wei, 2017*). Once metastatic foci are established in distant organs, the 5-year overall survival rate of melanoma patients drops sharply to less than 10% (*Braeuer et al., 2014*). Therefore, it is urgent to identify the mechanisms that drive the occurrence and metastasis of CM, and to develop effective therapeutic strategies. Studies had classified the somatic mutations and expression profiles of metastatic melanoma (*Cancer Genome Atlas Network, 2015*), but the mechanisms of CM evolution and metastasis have not been fully understood (*Shain et al., 2018*). Understanding the gene expression changes during the development of CM will help to develop new biomarkers and therapeutic targets for the diagnosis and treatment of patients.

Gene expression microarray technology can be used to understand the biology associated with cancers, gene mutations and abnormal biological pathways, as well as to predict the diagnosis, treatment, prognosis or metastasis of patients (*Tao et al., 2017*). The results of microarray technology provide a wealth of information; thus, the data stored in public databases can be reintegrated and bioinformatics analyzed to search for new clues about the pathological mechanisms of cancers through computers rather than laboratories (*Wei et al., 2019*). In recent years, a large number of studies have predicted the key genes, signaling pathways and protein functions (*Le et al., 2019*) of many cancers by analyzing the patients' genetic profiles. For example, the pathogenesis and metastasis mechanism of colorectal cancer (*Huang et al., 2018*), prostate cancer (*Wang et al., 2018*), breast cancer (*Bertucci et al., 2019*) and other cancers had been explored. A number of

studies have done bioinformatics analysis of CM. *Chen et al. (2020)* compared the DEGs between normal skin and melanoma, then used bioinformatics methods to analyze and identify the pathogenesis of CM. *Wang et al. (2020)* compared the expression of CD38 in the tissues of healthy people and melanoma patients in the TCGA database, and analyzed the occurrence of subtypes and promoter methylation, so as to conclude that CD38 may be a potential biomarker for CM. Some studies obtained differentially expressed non-coding RNAs by analyzing the microRNA and lncRNA of melanoma, which proposed more possibilities for its occurrence and development mechanism (*Li et al., 2019b*). Meanwhile, some articles have reported the metastasis of CM. For example, *Chen et al. (2019)* analyzed the gene expression of primary and metastatic melanoma in a database and obtained some candidate genes for metastasis. *Wang et al. (2019b)* comprehensively analyzed the gene expression of PM and MM in TCGA and constructed a competitive endogenous RNA (ceRNA) network, then proposed a new idea that non-coding RNA and mRNA may act together on the metastasis of melanoma.

However, previous studies have mainly focused on the analysis of PM and N samples. Several studies have also explored the metastasis of CM, but the entire progression of melanoma has not been analyzed from the perspective of occurrence and development, nor has it been compared by combining multiple data sets. In this study, we downloaded three gene expression profiles (GSE7553, GSE15605, and GSE46517) from the GEO, which all included N, PM and MM samples. DEGs among N, PM and MM were determined by gene expression profiling. Subsequently, GO functions, KEGG pathways and PPI network analyses were performed on DEGs. Finally, verification and survival analysis were performed on identified hub genes, which may be potential biomarkers and therapeutic targets in the occurrence and transfer of CM. The flow chart is shown in Fig. 1.

## MATERIALS & METHODS

### GEO gene expression data

Three gene expression datasets (GSE7553 (*Riker et al., 2008*), GSE15605 (*Raskin et al., 2013*), and GSE46517 (*Kabbarah et al., 2010*) were obtained from the GEO database (http://www.ncbi.nlm.nih.gov/geo). The file type of the original gene expression data set was CEL, and the platform file contained probe ID, gene marker and entrez gene ID. GSE7553 and GSE15605 were based on the GPL570 platform (Affymetrix Human Genome U133 Plus 2.0 Array) and GSE46517 was based on the GPL96 platform (Affymetrix Human Genome U133A Array). There were a total of 225 tissue samples in the three data sets, including 28 normal skin samples, 93 primary melanoma samples and 104 metastatic melanoma samples.

### Data processing and DEGs filtering

The raw CEL files were background-adjusted and standardized by the R software (*Smyth, 2004*). According to the annotation file, the probe ID was replaced with the corresponding gene symbol. If there were multiple probes for the same gene, the R language was used to calculate the average value for further analysis. Then the limma R package was used to screen the genes of each data set, when the $p$-value $<0.05$ and $|\log_2$ fold change (FC)$|> 1$

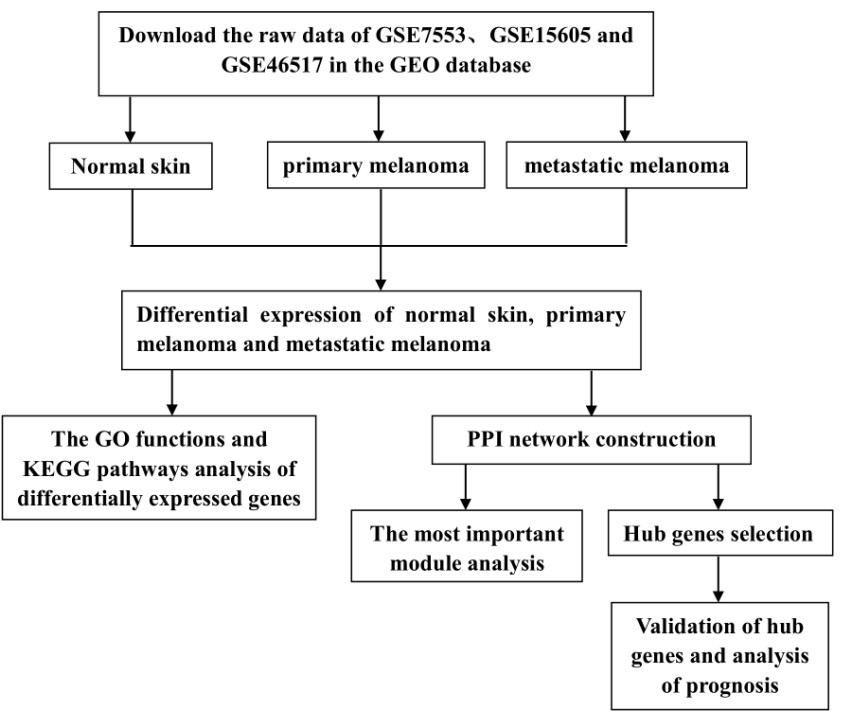

**Figure 1  Experimental flow chart.**

were considered DEGs (*Zhang et al., 2019*). The upregulated or downregulated DEGs lists were overlapped by Venn diagram (http://bioinformatics.psb.ugent.be/webtools/Venn/), for subsequent function analysis.

Using DAVID 6.8 database (https://david.ncifcrf.gov/home.jsp) to analyze the GO functions and KEGG pathways of integrated DEGs (*Xing et al., 2016*). The GO terms and the KEGG pathways with $p < 0.05$ were selected to be the enriched functions. GO functions analyses covered three domains: Biological Process, Cellular Component and Molecular Function.

## PPI network and the most important module analysis

The PPI network was constructed by the STRING (https://string-db.org/) platform, an online tool used for revealing protein interactions and functional analysis (*Szklarczyk et al., 2017*). In PPI network, each node represents a protein and each edge represents the action between proteins (*Khunlertgit & Yoon, 2016*). Then, the PPI network was visualized by Cytoscape software.

The most important module in PPI network was identified by means of the plug-in Molecular Complex Detection (MCODE) (*Li et al., 2017*). The criteria for selection were as follows: degree cut-off = 2, node score cut-off = 0.2, Max depth = 100, and k-score = 2. Subsequently, the GO functions and KEGG pathways analyses for genes in these modules were performed by DAVID, and $p < 0.05$ was considered statistically significant.

**Table 1  The information of GEO dataset and DEGs.**

| GEO datasets | Platform | Normal skin (N) | Primary melanoma (PM) | Metastatic melanoma (MM) | Differentially expressed genes (DEGs) | | |
|---|---|---|---|---|---|---|---|
| | | | | | PM and N | MM and N | MM and PM |
| GSE7553 | GPL570 | 4 | 16 | 40 | 843 | 1129 | 929 |
| GSE15605 | GPL570 | 16 | 46 | 12 | 2605 | 2829 | 1560 |
| GSE46517 | GPL96 | 8 | 31 | 52 | 668 | 813 | 2466 |

## Hub genes selection and analysis

Through 12 topological analysis methods, the cytoHubba of R software was used to sort the nodes in the PPI network. The hub genes consists of the overlapping results, which were obtained by the top 10 nodes of the Maximal Clique Centrality (MCC) analysis method and the degree of gene ≥10 (*Chin et al., 2014*). Subsequently, Pathway Commons Network Visualizer (PCViz), an open platform for exploring multidimensional cancer genome data, was used to analyze the association between hub genes and their co-expressed genes. The biological process analysis of hub genes was visualized by the Biological Networks Gene Oncology tool (BiNGO) plugin of Cytoscape (*Maere, Heymans & Kuiper, 2005*).

## Validation of hub genes and survival curve analysis

The UALCAN website (http://ualcan.path.uab.edu/) was used to analyze the TCGA gene expression data, in order to compare the expression of hub genes in normal skin, primary melanoma and metastatic melanoma samples (*Chandrashekar et al., 2017*). Then, the overall survival curve of each hub genes were analyzed by Gene Expression Profiling interactive analysis (GEPIA) (http://gepia.cancer-pku.cn/), and $p < 0.05$ was considered as a statistically significant difference (*Tang et al., 2017*).

# RESULTS

## Identification of DEGs

R software was used to compare the gene expression of samples from GSE7553, GSE15605 and GSE46517 data sets, and the DEGs of N, PM and MM were obtained in each data sets (Table 1, table s1-9, available at https://doi.org/10.6084/m9.figshare.13019600.v1). Then, the overlaps of 134, 317, and 147 DEGs between PM and N, MM and N, and MM and PM are obtained from the three data sets, which were shown by Venn disgram (Fig. 2). Among them, there were 12 upregulated genes and 122 downregulated genes in the PM compared to N (table s10, available at https://doi.org/10.6084/m9.figshare.13019600.v1), 153 upregulated genes and 164 downregulated genes between MM with N (table s11, available at https://doi.org/10.6084/m9.figshare.13019600.v1), and MM had 79 upregulated genes and 68 downregulated genes compared with PM (table s12, available at https://doi.org/10.6084/m9.figshare.13019600.v1).

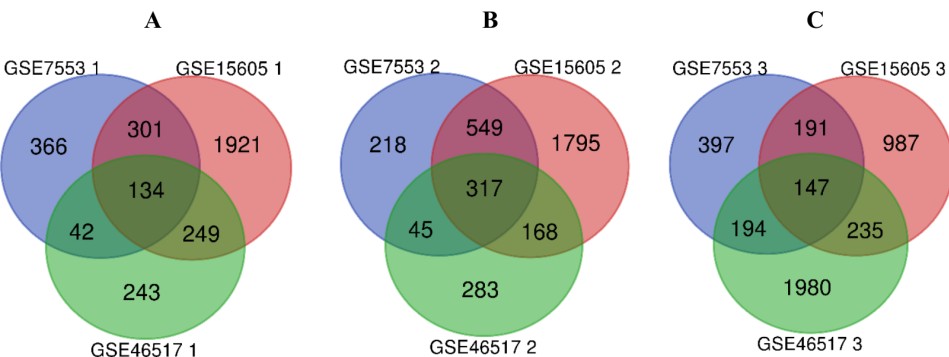

**Figure 2  Venn diagram of DEGs in GSE7553, GSE15605 and GSE46517.** According to |log$_2$ fold change (FC)| > 1 and $p$-value < 0.05, DEGs were identified along normal skin, primary cutaneous melanoma and metastatic cutaneous melanoma samples in GSE7553, GSE15605 and GSE46517 data sets. (A) The overlap of 134 DEGs was shown between the PM and N samples of the three data sets. (B) The overlap of 317 DEGs was shown between MM and N of the three data sets. (C) The overlap of 147 DEGs was shown between the MM and PM of the three data sets.

## GO functions and KEGG pathways enrichment analyses of DEGs

DAVID was used for GO functions and KEGG pathways enrichment analysis (Figs. 3–5, table s13-15, available at https://doi.org/10.6084/m9.figshare.13019600.v1). GO functions analysis results showed that compared with N samples, upregulated genes of PM were enriched in the collagen catabolic process (BP), while downregulated genes were enriched in transcription from RNA polymerase II promoter (BP), plasma membrane (CC) and structural molecule activity (MF). The upregulated genes between MM and N samples were enriched in negative regulation of neuron apoptotic process (BP), spindle microtubule (CC) and protein kinase activity (MF), and the downregulated genes were enriched in epidermal development (BP), extracellular exosome (CC) and structural molecule activity (MF). In MM and PM samples, the upregulated genes mainly included cell division (BP), cytoplasm (CC), and protein binding (MF), and the downregulated genes mainly included epidermis development (BP), extracellular exosomes (CC), and structural molecule activity (MF).

The KEGG pathways of the overlapped DEGs were analyzed, the upregulated genes between PM and N were significantly enriched in the transcriptional misregulation in cancer, while the downregulated genes were enriched in the metabolic pathways. In MM and N, the upregulated DEGs enriched in the pathway in cancer and the transcriptional misregulation in cancer, and the downregulated DEGs enriched in the arachidonic acid metabolism and steroid biosynthesis. Complement and coagulation cascades was the top enriched term for upregulation genes of MM and PM, while the p53 signaling pathway was the top enriched term for downregulation genes.

## PPI network construction and the most meaningful module analysis

The PPI network of DEGs was constructed using the STRING (Fig. 6) and the most important modules were obtained by Cytoscape (Fig. 7). The GO functions and KEGG pathways enrichment analysis showed that the important modules of PM and N were

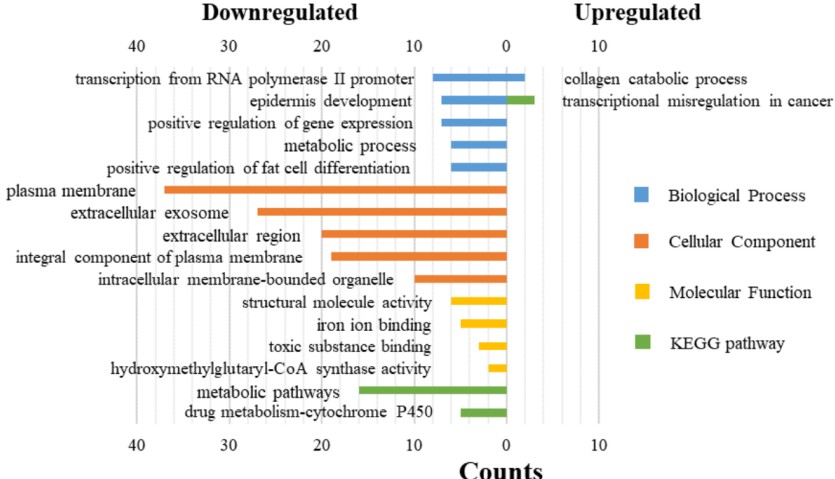

**Figure 3** GO functions and KEGG pathways enrichment analysis of the upregulated and downregulated genes between primary cutaneous melanoma and normal skin.

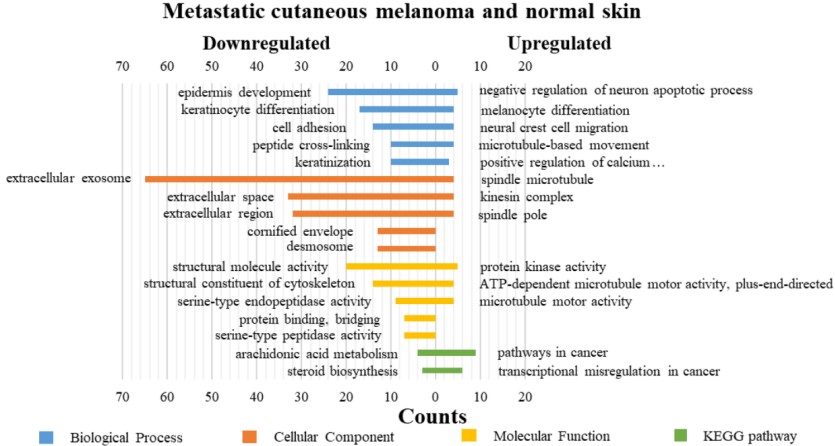

**Figure 4** GO functions and KEGG pathways enrichment analysis of the upregulated and downregulated genes between metastatic cutaneous melanoma and normal skin.

enriched in the cholesterol biosynthetic process, mitochondrion, oxidoreductase activity and metabolic pathway (Fig. 8A, table s16, available at https://doi.org/10.6084/m9.figshare.13019600.v1). The module genes between MM and N were mainly enriched in cell division, extracellular exosomes, protein binding and oocyte meiosis pathway (Fig. 8B, table s17, available at https://doi.org/10.6084/m9.figshare.13019600.v1). In the MM and PM, the module genes were enriched in keratinocyte differentiation, cytoplasm, structural molecule activity, protein binding and mismatch repair (Fig. 8C, table s18, available at https://doi.org/10.6084/m9.figshare.13019600.v1).

**Figure 5** GO functions and KEGG pathways enrichment analysis of the upregulated and downregulated genes between metastatic cutaneous melanoma and primary cutaneous melanoma.

## Hub gene selection and analysis

According to the above criteria, 14, 18 and 18 genes among N, PM and MM were selected as the hub genes in PPI network, and the details are shown in Table 2. PCViz online platform was used to construct the hub genes and their co-expressed genes network (Fig. s1, available at https://doi.org/10.6084/m9.figshare.13019600.v1). The biological process analysis of hub genes was shown in figure s2 (available at https://doi.org/10.6084/m9.figshare.13019600.v1).

## Validation of hub genes and survival curve analysis

The transcription expression data of hub genes from 473 TCGA samples were analyzed using UALCAN. Among them, 1 case was normal sample, 104 cases were PM samples, and 368 cases were MM samples. We found that the expression of those hub genes in MM samples decreased significantly compared with PM samples (Fig. 9). Therefore, the results of the candidate hub genes identified by us are reliable.

We utilized the GEPIA online tool to analyze the samples data from TCGA and obtain the overall survival curve of these hub genes in skin melanoma patients, so as to further study the relationship between hub genes and patient survival and prognosis. As shown in Fig. 10, the changes of IVL, FLG, SPRR1B, DSG3, KRT5, DSG1, KRT16, PKP1, KRT14 and DSC3 in melanoma patients were associated with shortened overall survival, which suggested that these hub genes expression differences may be related to the progression and prognosis of cutaneous melanoma, thus can be used for predicting the deterioration and improvement of CM.

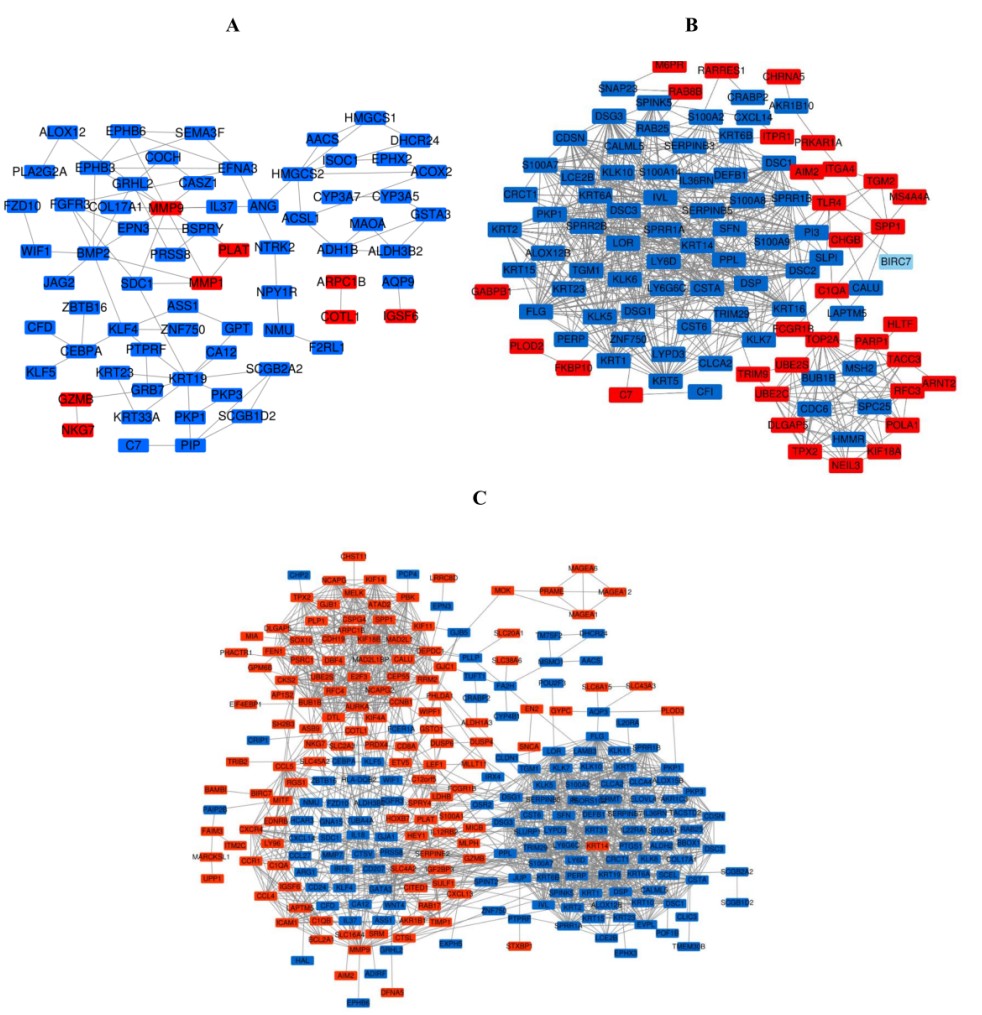

**Figure 6** **The PPI network of DEGs was constructed by using Cytoscape.** (A) PPI network of DEGs between PM and N of the three data sets. (B) PPI network of DEGs between MM and N of the three data sets. (C) PPI network of DEGs between MM and PM of the three data sets.

## DISCUSSION

Recently, many studies had carried out gene expression profiling and bioinformatics analysis on the molecular mechanism of CM occurrence, but the biological mechanism of its development and metastasis were still unclear. In this study, we downloaded three gene expression data sets from GEO and used a comprehensive bioinformatics method to directly compare the gene expression differences among N, PM and MM samples. 134, 317 and 147 DEGs, as well as 14, 18, and 18 hub genes were identified between PM and N, MM and N, MM and PM, respectively. Then, we used the online analysis website to verify the hub genes expression in TCGA samples and performed survival analysis on CM patients.

Through GO functions and KEGG pathways analyses of DEGs, we found that biological processes of upregulated genes mainly concentrated in cell division, spindle microtubule,

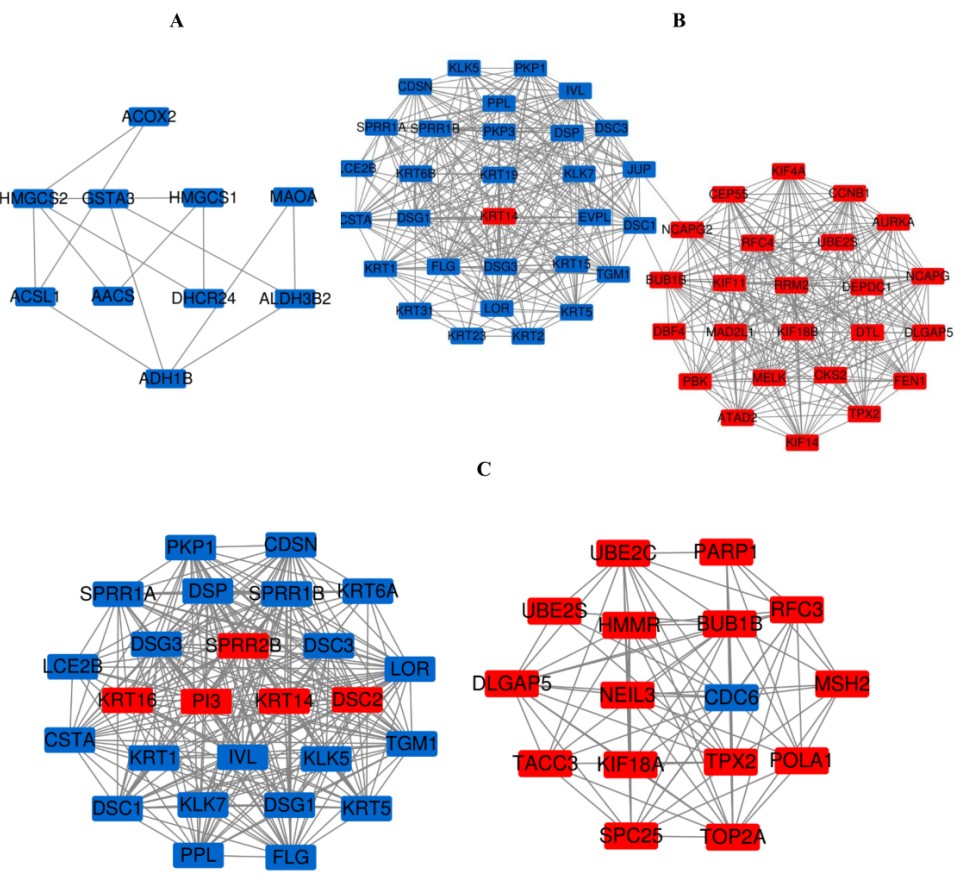

**Figure 7 The most meaningful module from the PPI network.** Upregulated gene marker is light red, downregulated gene marker is light blue. (A) The most significant module of PM and N. (B) The most significant module of MM and N. (C) The most significant module of MM and PM.

protein kinase activity and the pathway of transcriptional misregulation in cancer. The downregulated gene mainly occurred in epidermis development, extracellular exosome, structural molecule activity, metabolic pathways and p53 signaling pathway. Studies have shown that the occurrence and metastasis of melanoma need to be realized through the promotion of cell mitosis and the growth of anti-aging and anti-apoptosis (*Bennett, 2016*). In addition, spindle microtubules can accelerate the proliferation and transfer of cells, and the regulation of metabolic pathways such as protein synthesis and transcriptional disorders can promote cell division (*Soltani et al., 2005*). The results showed that downregulated genes were associated with skin epidermal development, melanoma cells were produced in the basal layer of the epidermis and hair follicles, and epidermal keratinization could control the homeostasis of melanocytes (*Orgaz & Sanz-Moreno, 2013*). Studies have found that most exosomes mediate the tumor process in the progression of melanoma (*Mannavola et al., 2019*). Therefore, previous studies had confirmed our results.

In the 14 hub genes between PM and N, it had been found that matrix metalloproteinases (MMP) can participate in the skin matrix remodeling through degrading and rebuilding

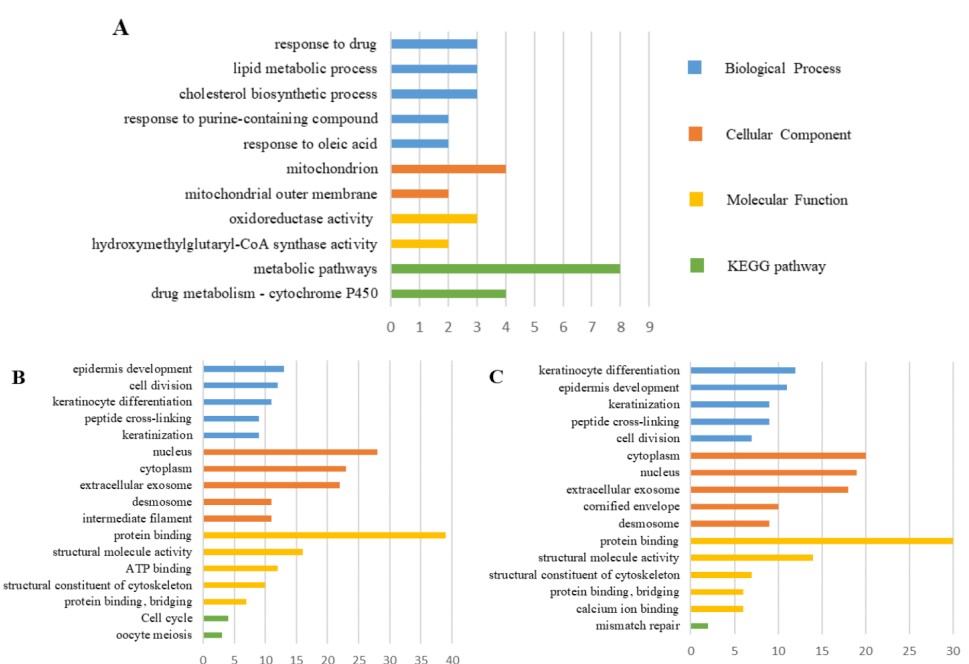

**Figure 8** GO functions and KEGG pathways enrichment analysis of the modular genes among normal skin, primary cutaneous melanoma and metastatic cutaneous melanoma.

**Table 2** The hub genes among normal skin, primary cutaneous melanoma and metastatic cutaneous melanoma samples.

|  | PM and N | MM and N | MM and PM |
|---|---|---|---|
| Upregulated | MMP9, MMP1 | AURKA, CCNB1, TPX2 |  |
| Downregulated | KRT19, BMP2, HMGCS2, KLF4, EPHB3, EFNA3, PIP, ADH1B, FGFR3, SDC1, CEBPA, EPHB6 | LOR, FLG, JUP, DSC1, SPRR1B, DSG1, DSP, SPRR1A, BUB1B, KRT5, KRT14, CDSN, IVL, DSG3, TGM1 | LOR, IVL, FLG, SPRR1B, DSG3, KRT5, CDSN, TGM1, DSG1, KRT16, SPRR1A, PKP1, KRT14, DSC3, DSP, CSTA, S100A7, DSC1 |

the matrix components, and affect the proliferation, survival, vascularization, protease expression and migration of melanoma cells (*Napoli et al., 2020*). MMP-9 knockdown can reduce the migration and invasion of melanoma cells and inhibit epithelial-mesenchymal transformation (EMT), thus being considered as a promising molecule for the CM treatment (*Tian et al., 2019*). Meanwhile, bone morphogenetic protein (BMPs) is involved in the regulation of MMPs and is an inevitable factor in the migration and invasion of melanoma cells (*Rothhammer, Braig & Bosserhoff, 2008*). Fibroblast growth factor receptor 3 (FGFR3) may promote melanoma growth, metastasis, and EMT behavior by influencing the phosphorylation levels of ERK, AKT, and EGFR (*Li et al., 2019a*). The loss of EphB6 may have deleterious immunological effects in cancer progression, while *Hafner et al. (2003)* found that its expression decreased gradually in N, PM and MM. These previous studies had suggested that these hub genes may have a potential role in the development of CM.

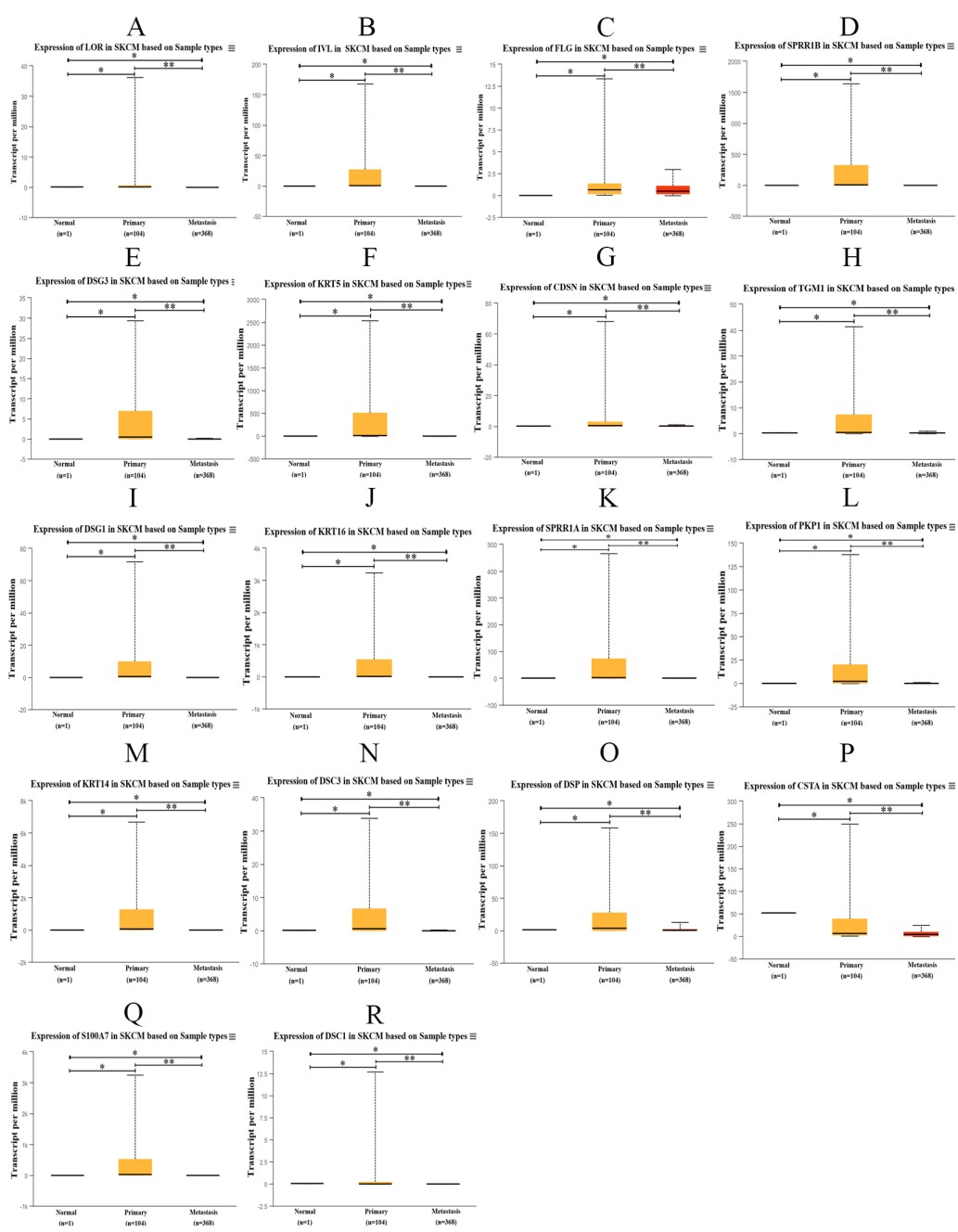

**Figure 9 Box plot of gene expression values for hub genes in normal skin, primary and metastatic cutaneous melanoma samples.** (A) LOR (B) IVL (C) FLG (D) SPRR1B (E) DSG3 (F) KRT5 (G) CDSN (H) TGM1 (I) DSG1 (J) KRT16 (K) SPRR1A (L) PKP1 (M) KRT14 (N) DSC3 (O) DSP (P) CSTA (Q) S100A7 (R) DSC1, $p < 0.05$ was considered statistically significant. (* $p > 0.05$, ** $p < 0.05$).

In MM and PM, we found 18 hub genes through PPI network, all of which were downregulated genes. Several were associated with keratinocyte differentiation and epidermal development, such as loricrin (LOR), involucrin (IVL), filaggrin (FLG), small

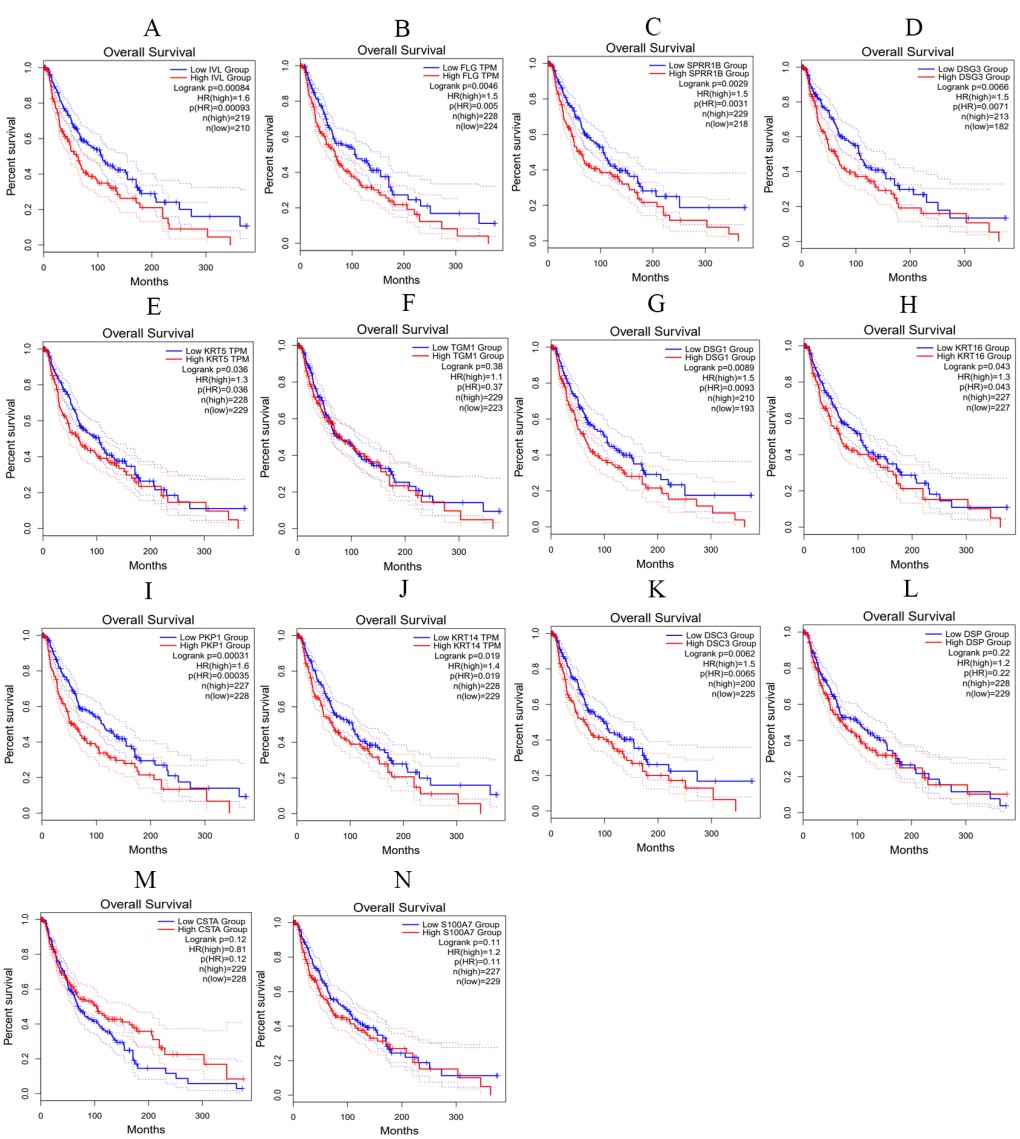

**Figure 10** **The overall survival curves of hub genes in TCGA database.** (A) IVL (B) FLG (C) SPRR1B (D) DSG3 (E) KRT5 (F) TGM1 (G) DSG1 (H) KRT16 (I) PKP1 (J) KRT14 (K) DSC3 (L) DSP (M) CSTA (N) S100A7, *p* < 0.05 was considered statistically significant.

proline-rich protein1 (SPRR1) keratin (KRT) and plakophilin (PKP1), which may result in loss of epidermal function. Among them, some hub genes had been found to be related to the production and metastasis of melanoma, such as SPRR1 (*Koizumi et al., 1996*) and PKP1 (*Wang et al., 2019a*). However, other genes had not been proven to be related to CM, such as LOR dysregulation was considered as an early indicator of potential malignant diseases, including oral submucosal fibrosis and leucoplasis (*Nithya et al., 2019*). IVL was a specific and sensitive marker of cell differentiation, the expression of IVL in head and neck squamous cell carcinoma patients with or without lymph node metastasis was significantly different (*Jin & Qin, 2020*). Loss-of-function mutations in FLG can lead to a

decrease in epidermal filaggrin and its degradation products, and increase the sensitivity of CM (*Thyssen et al., 2018*). In MM and PM, the downregulated amplitude of FLG ($\log_2$ FC = −5.404) was less than the amplitude in MM and N ($\log_2$ FC = −8.586), while there was no significant difference between PM and N. It showed that FLG may be related to CM transfer, and this association needs to be verified by subsequent experiments. Studies had shown that KRT5 and KRT14 were involved in HNSCC differentiation and apoptosis as the epithelial proliferative markers (*Wang et al., 2014*). Meanwhile, immunohistochemical staining of KRT14 and KRT16 in PM and MM were mostly negative, and the positive distribution contributed to the diagnosis of poorly differentiated squamous cell carcinoma (*Safadi et al., 2016*). CDSN was expressed in hair follicles and keratinized epithelial cells, played an important role in intercellular adhesion, and was related to skin barrier function and epidermal defense pathway (*Mondon et al., 2017*). Studies had found that mutations in the CDSN gene could cause excessive keratosis of the skin, and lead to peeling skin disease and hypotrichosis simplex of the scalp (*Van der Velden et al., 2020*), therefore, down-regulation of CDSN may accelerate the development of CM by slowing down epidermal development.

In MM and PM, there were some hub genes associated with EMT. EMT was normally associated with embryogenesis and wound healing, but in tumor cells, it promoted tumor metastasis by enabling cells to leave the epithelium and acquire mesenchymal specificity (*Hodorogea et al., 2019*). This process increased the aggressiveness of the tumor by the loss of the epithelial phenotype (E-cadherin, desmosin, laminin-1) and the acquisition of the mesenchymal marker (N-cadherin) (*Kalluri & Robert, 2009*). Hub gene desmoglein1(DSG1) could control the role of keratinocytes, and contribute to the page-like behavior in the development of melanoma (*Arnette et al., 2020*). Furthermore, desmocollin1(DSC1) and desmocollin3(DSC3) are members of the E-cadherin superfamily, involved in cell–cell adhesion and cell-extracellular matrix interaction. Benign melanocytes expressed high levels of E-cadherin, and during the transition to melanoma cells, E-cadherin was down-regulated and N-cadherin was up-regulated (*Jaeger et al., 2007*). Studies had shown that desmoplakin (DSP) was a desmosomal protein involved in cell–cell adhesion. Desmosome formation was characteristic of cell differentiation and intercellular adhesion, and the loss of desmosome might accelerate the occurrence and early migration of tumor cells (*Walia et al., 2014*).

There are also some hub genes between MM and PM had been found to be related to CM, for example, the down-regulation of cystatin A (CSTA) expression has become an important feature to distinguish N, PM and MM (*Wallin et al., 2017*). Several S100 family genes had been found to be highly expressed in PM, but low level in MM (*Xiong, Pan & Li, 2019*). In particular, the loss of S100A7 is highly correlated with the metastasis progression score (*Bhalla et al., 2019*). *Lentini et al. (2008)* observed that the anti-invasion effect of transglutaminase(TGM) might lead to the post-translational modification of some components of the cell basal membrane, thereby interfering with the metastasis of melanoma cells. The activity of TGM2 had a protective effect on the progression of melanoma in vivo (*Facchiano et al., 2013*), but no studies had been conducted to prove the relationship between TGM1 and CM, so TGM1 may be a new potential marker.

We found that most of the above hub genes had been reported to be closely related to the generation and metastasis of CM. Moreover, through prognostic analysis, most hub gene expression differences in CM patients were connected with overall survival, which proved the reliability of our study. There are still a few genes that had not been reported or experimentally confirmed to be associated with CM, but some of them are related to the occurrence and development of other cancers. So they might be potential biomarkers of CM, and a large number of experiments are needed to confirm. Most hub genes, such as LOR, IVL, FLG, DSG3, TGM1, KRT16, SPRR1A, KRT14, DSP and CSTA, showed no difference in the expression of PM and N, but significantly decreased in MM and PM, suggesting that these genes might be potential predictors of CM metastasis. The expression of some genes, such as CDSN, DSG1, DSC3, DSC1 and DSP, was downregulated in all three groups, which might be relate with the occurrence and progression of CM. It is worth noting that three genes, SPRR1B, PKP1 and S100A7, were upregulated in PM and N, but downregulated in MM and PM, which were likely to be used as novel markers to distinguish whether CM was metastatic or not. These assumptions need to be tested experimentally.

Compared with previous studies, we used more samples, and compared N, PM and MM in pairs, then took the intersection, so as to make the experimental results more reliable. Besides using GEO data sets, we also used TCGA data for verification, which increased the sample size and accuracy. However, the limitation of this study is the lack of experimental verification of hub genes. Therefore, to understand whether the hub genes are really closely related to the generation and metastasis of CM, a large number of subsequent experiments are needed to explore.

## CONCLUSIONS

In conclusion, through bioinformatics analysis, we obtained some potential key genes and biological pathways related to the occurrence and metastasis of CM, providing directions for future research. In addition to the genes that have been reported and demonstrated, other identified key genes may be potential prognostic markers and therapeutic targets for CM occurrence and metastasis.

## ACKNOWLEDGEMENTS

We would like to thank all teachers in Department of Laboratory Medicine for their thoughtful kindness.

### Funding

This work was supported by the National Natural Science Funding of China (No.81273002 and No.81471499) and the Hunan Provincial Natural Science Foundation of China (No.2019JJ40347). There was no additional external funding received for this study. The funders had no role in study design, data collection and analysis, decision to publish, or preparation of the manuscript.

## Grant Disclosures

The following grant information was disclosed by the authors:
The National Natural Science Funding of China: 81273002, 81471499.
Hunan Provincial Natural Science Foundation of China: 2019JJ40347.

## Competing Interests

The authors declare there are no competing interests.

## Author Contributions

- Hanying Dai conceived and designed the experiments, performed the experiments, analyzed the data, prepared figures and/or tables, authored or reviewed drafts of the paper, and approved the final draft.
- Lihuang Guo conceived and designed the experiments, analyzed the data, authored or reviewed drafts of the paper, and approved the final draft.
- Mingyue Lin, Zhenbo Cheng, Jiancheng Li, Jinxia Tang, Xisha Huan and Yue Huang conceived and designed the experiments, authored or reviewed drafts of the paper, and approved the final draft.
- Keqian Xu conceived and designed the experiments, performed the experiments, analyzed the data, authored or reviewed drafts of the paper, and approved the final draft.

## Data Availability

Raw data are available at figshare: Dai, Hanying (2020): Supplemental files. figshare. Online resource. https://doi.org/10.6084/m9.figshare.13019600.v1.

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
