# Peer review of "Comprehensive analysis and identification of key genes and signaling pathways in the occurrence and metastasis of cutaneous melanoma"

_PeerJ, doi:10.7717/peerj.10265_

## Round 0.1 · original submission · Major Revisions

As you will see, the reviewers consider your Ms needs major changes and reformulations. Please address all the reviewers' concerns in the revised version of your work.

·

Basic reporting

(1) The language in this manuscript is ambiguous, and the conclusion is exaggerated. Please see Major Q1.
(2) Picture quality is basically no problem..
(3) No raw data is provided.
(4) Literature well referenced.

Experimental design

(1) The experimental method has some problems and needs to be redesigned. Please see Major Q2 and Q3.
(2) There is no problem with the statistical method in this manuscript.

Validity of the findings

(1) The findings of this study need further experiments. Please see Major Q1 to Q3.

Additional comments

Melanoma is a malignant tumor of melanocytes, and the incidence has increased faster than any other cancer over the past half century. Most primary cutaneous melanoma can be cured by local excision, but metastatic cutaneous melanoma has a poor prognosis. Therefore, the research on the mechanism of cutaneous melanoma metastasis will be beneficial to diagnose the metastatic melanoma early, improve the treatment, and prolong the survival of patients. In this study, Dai Hangying and colleagues identified the key genes and molecular pathways involved in the occurrence and metastasis of cutaneous melanoma. The study provides the conclusion, which showed that some DEGs were found from normal skin, primary melanoma and metastatic melanoma samples. These hub genes may play an important role in the generation, invasion, recurrence or death of CM, providing new ideas and targets for the diagnosis and treatment of metastatic cutaneous melanoma. There are some problems in this study, which require the author to revise the research design, and there are some major concerns that need to be addressed.

Major concerns:
1. “Validity of the findings” The title of this manuscript is that “Comprehensive analysis and identification of key genes and signaling pathways in cutaneous melanoma metastasis”. However, from the analysis results and conclusions of the manuscript, the author did not identify convincing analysis, but only found a bunch of differentially expressed genes, analyzed the potential functions and pathways, and found a dozen genes related to prognosis. The correlation analysis used in this study can only indicate the possibility of an important role and should not be exaggerated. The author had to make a careful revision of the language.
2. “Validity of the findings” The authors used multiple databases to analyze differentially expressed genes between primary melanoma and metastatic melanoma and identified these genes as the key genes associated with melanoma metastasis. However, the process of metastasis is complex, and the genes specifically expressed in the metastasis foci may be a passive process for the cancer cells to adapt to the microenvironment of the metastasis foci, rather than "drivers". Therefore, there is a big deviation between the subject design idea and the research objective in this paper, and the results and conclusions cannot support each other. I would suggest that the authors reformulate the purpose of the study and redesign the analysis process according to the purpose. The key is not to make the purpose too large, but to focus on a specific clinical problem or tumor biology problem.
3. “Results – Validation of hub genes and survival curve analysis” Line 201-208. When using TCGA database for verification, the author firstly verified the expression differences of selected genes in primary foci (n = 104) and metastatic foci (n = 368), and then analyzed the relationship between related genes and survival prognosis of patients, proving the importance of related genes. However, as can be seen from the survival curve in Figure 8, the author did not simply analyze the relationship between the expression of relevant genes in the primary site and the survival prognosis of patients, but included all the data of the primary site and the metastatic site into the analysis (n = 250+), which was obviously a logical problem. The prognosis of patients with metastatic foci is certainly worse than that of patients with primary foci alone. The analyzed genes have been found to be abnormally expressed in metastatic foci. If the samples of all patients are integrated and analyzed, the correlation between these genes and prognosis is inevitable and has no real diagnostic value. Therefore, the authors analyzed the survival or prognosis of related genes in patients with primary and metastatic foci respectively.

·

Basic reporting

The authors provide a detailed bioinformatics analysis of the mechanisms involved in cutaneous melanoma progression. I believe that some aspects of their analyses are over-represented, while some other ones are not. I would focus mainly on the comparison between PM and MM.
I beleive that the English language should be improved. Please check that the correct tense for all verbs is used .

Experimental design

My major concern regarding the experimental design is whether the comparison between melanoma (primary or not) and normal skin, in order to identify Differentially Expressed Genes, is valid. I mean that the comparison between cancer and normal melanocytes could be much more meaningful.Could the authors check also this comparison at least at the datasets that contain samples of normal melanocytes or nevi?
I would suggest that the authors include a table describing the datasets that they have analysed, presenting details concerning the number of normal skin, primary or metastatic samples per analysis. The number of differentiated genes could also be indicated.

Validity of the findings

I believe that the presented bioinformatics analyses are robust and statistically sound.
In my opinion there are many tables concerning the GO and Kegg-based analyses. I would suggest to keep those concerning the PM versus MM DEGs and present the others as supplementary files.
Regarding figure 7 I couldn’t understand why the y axis is indicated in Transcripts per million and why the category normal is presented with n=1.

Additional comments

Please correct lines 271-272

Reviewer 3 ·

Basic reporting

- The use of English is verbose and not clear. There also contains grammatical errors and typos. The authors should re-check and revise carefully.
- Literature reviews are weak. The authors should add more literature references about some related works.
- Some abbreviations need to be defined at the first use.

Experimental design

- Research design is a big concern. Why did the authors perform the analysis among three groups? If the problem aims to address cutaneous melanoma metastasis, why did the authors not compare between metastasis and non-metastasis only? Or if the authors compared three groups, why did the authors separate into three binary problems? Is it possible if the authors perform the analysis on all three groups together?
- Methods have not been explained well and it is not easy to replicate. Also, it is important that the authors could release source codes for analysis.
- GO database and analysis has been used in previous works related to biomedical such as PMID: 31277574 and PMID: 31921391. Therefore, the authors should provide some references in this description.

Validity of the findings

- The choice of p-value and cut-off value is a question. Why did sometimes the authors select significant level of 0.01, 0.02, or 0.05 etc? At least, these values must be consistent.
- There are many works that have addressed the same question such as PMID: 31173190, PMID: 31937175, or PMID: 32547879. Thus what are differences between this study and the others? The authors should discuss and compare with some related works.

Additional comments

No comment

---

## Round 0.2 · Minor Revisions

Although the authors have addressed the majority of the reviewers' concerns, some improvements are needed in the manuscript. Please address all the points raised by the reviewer.

In addition:

- In the abstract please include more precise information about the pathways deregulated in primary Melanoma vs NT vs Metastatic lesions.

- In line 202 please confirm the number of cases in the sentence "Among them, 1 case was normal sample, 104 cases were PM samples, and 368 cases were MM samples. We found that the expression of those hub genes in MM samples decreased significantly compared with PM samples (Fig. 11)." The series only includes a single normal sample and 104 primary melanomas?

- In line 235 please delete the sentence "The inactivation of p53 has an influence on the occurrence of melanoma[36], and the defective p53 pathway also has an anti-apoptotic effect[37].". It is incorrect!

- As noted by the reviewer, the manuscript has an excessive number of Figures. Please reduce the number of Figures or transfer some to Supplementary data.

·

Basic reporting

No comment.

Experimental design

No comment.

Validity of the findings

No comment.

Additional comments

Melanoma is a malignant tumor of melanocytes, and the incidence has increased faster than any other cancer over the past half century. Most primary cutaneous melanoma can be cured by local excision, but metastatic cutaneous melanoma has a poor prognosis. Therefore, the research on the mechanism of cutaneous melanoma metastasis will be beneficial to diagnose the metastatic melanoma early, improve the treatment, and prolong the survival of patients. In this study, Dai Hangying and colleagues identified the key genes and molecular pathways involved in the occurrence and metastasis of cutaneous melanoma. The study provides the conclusion, which showed that some DEGs were found from normal skin, primary melanoma and metastatic melanoma samples. These hub genes may play an important role in the generation, invasion, recurrence or death of CM, providing new ideas and targets for the diagnosis and treatment of metastatic cutaneous melanoma. The paper is improved and most concerned raised by the reviewer have been addressed. But there are still questions to be answered.

Major concerns:
1. Too many pictures need to be deleted, too many network pictures, lack of presentation meaning.
2. Make sure the text is clearly visible in the figures.

Minor concerns:
1. Figure 11. Please convert the data logarithmically.

Reviewer 3 ·

Basic reporting

No comment

Experimental design

No comment

Validity of the findings

No comment

Additional comments

My previous comments have been addressed.

---

## Round 0.3 · accepted · Accept

Thank you for the modifications done in your manuscript and congratulations.